# Liupao Tea Extract Alleviates Rheumatoid Arthritis in Mice by Regulating the Gut–Joint Axis Mediated via Fatty Acid Metabolism

**DOI:** 10.3390/foods14162854

**Published:** 2025-08-18

**Authors:** Ying Tong, Zhiyong She, Xueting Lin, Jichu Luo, Xuan Guan, Mingsen Wen, Li Huang, Bao Yang, Xiaoying Liang, Song Xu, Yuru Tan, Pingchuan Zhu, Zhaoyang Wei, Haidan Liu, Xiadan Liu, Qisong Zhang

**Affiliations:** 1Guangxi Key Laboratory of Special Biomedicine, School of Medicine, Guangxi University, Nanning 530004, China; tongying0206@163.com (Y.T.); shezhiyong2023@163.com (Z.S.); kyl05170606@163.com (X.L.); luojichu1999@163.com (J.L.); guanxuan010203@163.com (X.G.); q954637097@163.com (M.W.); liangcai52711@163.com (X.L.); fighting68_xusong@163.com (S.X.); 15894549320@163.com (Y.T.); wzykyd@163.com (Z.W.); 13557257712@163.com (H.L.); 15755614946@163.com (X.L.); 2College of Light Industry and Food Engineering, Guangxi University, Nanning 530004, China; huangli1126@gxu.edu.cn; 3Hubei Provincial Key Laboratory of Occurrence and Intervention of Rheumatic Diseases, Hubei Minzu University, Enshi 445000, China; ybsept@hbmzu.edu.cn; 4State Key Laboratory for Conservation and Utilization of Subtropical Agro-Bioresources, Guangxi University, Nanning 530004, China; zhupingchuan12@163.com; 5Center for Instrumental Analysis, Guangxi University, Nanning 530004, China

**Keywords:** liupao tea, rheumatoid arthritis, gut microbiota, gut–joint axis, short-chain fatty acids, arachidonic acid metabolism

## Abstract

As a highly disabling chronic inflammatory disease, rheumatoid arthritis (RA) necessitates novel interventions. Liupao tea is a traditional Chinese dark tea known for its favorable anti-inflammatory properties. This study aims to elucidate the active ingredients and action mechanisms underlying the therapeutic effects of Liupao tea extract (LPTE) in RA. LPTE was preliminarily characterized by LC-MS technology. Network pharmacology and molecular docking predicted anti-RA compounds, targets, and pathways, with key compounds identified using chemical standards. The effect of LPTE on the collagen-induced arthritis mouse model was evaluated through serum biochemical analysis, micro-CT imaging, and histopathological analyses. Integrated serum metabolomics, 16S rRNA sequencing, MetOrigin analysis, SCFA metabolomics, and quantitative real-time PCR elucidated gut–joint axis mechanisms. LPTE effectively attenuated RA symptoms by reducing bone destruction and joint inflammation. Notably, LPTE reshaped gut microbiota by enriching key families such as *Monoglobaceae*, *Eggerthellaceae*, and *Desulfovibrionaceae*, thereby promoting SCFA production. Increased SCFA levels enhanced intestinal barrier integrity and exerted joint-protective and anti-inflammatory effects by upregulating tight junction proteins and activating SCFA receptors. LPTE also modulated arachidonic acid metabolism by affecting key genes such as *Alox5*, *Ptgs2*, and *Cbr1*. These effects collectively reduced the levels of pro-inflammatory cytokines and increased the expression of anti-inflammatory cytokines in joints. Additionally, quercetin, luteolin, ellagic acid, and kaempferol were identified as major anti-RA bioactive compounds in LPTE. Taken together, this study provides preliminary evidence that LPTE mitigates RA by regulating the gut–joint axis mediated via fatty acid metabolism.

## 1. Introduction

Rheumatoid arthritis (RA) is a severe chronic systemic inflammatory disorder affecting approximately 1% of the global adult population, posing a significant public health challenge [1]. RA is driven by a combination of genetic factors and environmental triggers, resulting in systemic inflammation, synovial hyperplasia, irreversible bone destruction, and cartilage erosion, which significantly increase the risk of disability and reduce quality of life [2,3]. RA progression is orchestrated by a complex cytokine network that activates relevant immune cells, promotes their proliferation and accumulation in the synovium, thereby perpetuating a vicious cycle of inflammation [4]. Additionally, arachidonic acid metabolism plays a pivotal role in RA pathogenesis, driving the production of pro-inflammatory cytokines and eicosanoids (prostaglandins, thromboxanes, and leukotrienes), which serve as key mediators of RA-associated inflammation [5,6]. Current drugs for treating RA have limitations, including reduced efficacy due to single targets and serious adverse reactions caused by long-term use [7,8]. Consequently, exploring safe and effective natural anti-inflammatory products to modulate cytokine production through multiple pathways represents a promising strategy for RA management.

Tea, derived from *Camellia sinensis* (L.) Kuntze, is the second most consumed non-alcoholic beverage globally, following water, with over three billion consumers across more than 160 countries [9]. Liupao tea (LPT), a fermented dark tea with a history spanning over 1500 years from Wuzhou, Guangxi, has gained popularity for its unique aroma, smooth taste, low bitterness, and recognized health benefits [10]. Rich in bioactive compounds such as polyphenols, catechins, and alkaloids, LPT undergoes microbial fermentation, which enhances the bioactivity of these compounds, particularly their antioxidant and anti-inflammatory properties [11,12]. Polyphenols in LPT have been demonstrated to significantly improve antioxidant activity and suppress the production of inflammatory factors, such as IL-6, IL-12, TNF-α, and IFN-γ [13,14]. Recent studies have indicated that LPT extracts, rich in catechins and alkaloids, restore oral mucosal barriers, promote tissue regeneration, and reduce microinflammation [15]. Additionally, LPT has been found to modulate gut microbiota, downregulate pro-inflammatory cytokines, and mitigate airway inflammation in allergic asthma [16]. These findings underscore the anti-inflammatory potential of LPT, indicating its promise as a complementary therapeutic approach for RA.

Emerging evidence highlights the critical role of gut microbiota dysbiosis in RA, establishing a strong link between gut health, systemic inflammation, and RA progression [17]. Gut dysbiosis impairs intestinal barrier function, facilitates antigen translocation, and increases autoimmune susceptibility, leading to bone remodeling defects and systemic inflammation [18,19]. The gut–joint axis represents a bidirectional communication network between the gut and joints, primarily mediated by gut-derived metabolites [20,21]. Short-chain fatty acids (SCFAs), gut microbiota-derived metabolites, inhibit joint inflammation and maintain gut–joint axis homeostasis by strengthening intestinal barrier integrity while downregulating pro-inflammatory cytokines (e.g., TNF-α and IL-6), and upregulating anti-inflammatory cytokines (e.g., IL-10) [17,22,23]. Excessive bone resorption by osteoclasts results in pathological bone loss, while certain SCFAs influence bone metabolism by regulating bone mass and preventing further bone loss in RA [24,25]. Interestingly, LPT produces probiotic-like effects by increasing the SCFA levels and the abundance of related gut bacteria [26]. Moreover, our previous research has also demonstrated LPT’s therapeutic effects on non-alcoholic fatty liver disease and hyperlipidemia by modulating the gut microbiota [11,27]. Collectively, these findings establish a rationale for investigating the therapeutic potential of LPT in RA through modulation of the gut–joint axis mediated via fatty acid metabolism.

Recent advances in bioinformatics and multiomics have enabled comprehensive investigations into the pharmacological properties and nutritional value of natural products and functional foods [11,27]. Network pharmacology is increasingly applied to identify bioactive ingredients and therapeutic targets in traditional Chinese medicine and functional foods, often integrated with molecular docking to simulate drug–target interactions [28]. Metabolomics provides a dynamic and systematic assessment of endogenous metabolic changes, elucidating the effects of disease treatments [29]. Gut microbiome analysis reveals alterations in microbial composition and abundance, along with their impacts on host metabolism [30]. Bioinformatics tools such as MetOrigin integrate metabolomics and gut microbiome data to elucidate microbial influences on metabolic pathways [11].

Given its anti-inflammatory and gut-modulating properties, LPT holds promising but underexplored potential for RA intervention by the gut–joint axis. This study was designed to systematically investigate the therapeutic effects of Liupao tea extract (LPTE) on RA by exploring the contribution of potential active ingredients and elucidating the underlying mechanisms, with a particular focus on its modulation of the gut–joint axis mediated via fatty acid metabolism. This research seeks to provide a scientific rationale for utilizing LPT in the prevention and management of RA and expand our understanding of its broader health-promoting properties.

## 2. Materials and Methods

### 2.1. Materials and Reagents

MS-grade acetonitrile, water, and formic acid were purchased from Merck & Co. (Billerica, MA, USA). Tamoxifen and cholic acid-d4 were obtained from Sigma-Aldrich Corporation (St Louis, MO, USA). Chicken type II collagen, complete and incomplete Freund’s adjuvant were sourced from Chondrex (Redmond, WA, USA). LPT was provided by China Tea Co., Ltd. (Wuzhou, China) (Batch number: S002/2022), which conformed to the requisitions of GB/T 32719.4-2016 (Brick LPT Second class). Methotrexate (MTX) was acquired from Shanghai Macklin Biochemical Co., Ltd. (Shanghai, China). Standard reference compounds (quercetin, ellagic acid, luteolin, caffeine, gallic acid, and kaempferol; HPLC > 99%) were purchased from Chengdu RefMedic Biotech Co., Ltd. (Chengdu, China). All other chemicals were of analytical reagent grade or higher.

### 2.2. Preparation and Characterization of LPTE

LPTE was prepared following previously reported methods with some modifications [31]. LPTE was prepared by aqueous extraction of pulverized LPT (1:10, *w*/*v*) at 100 °C for 1 h, repeated three times. Combined filtrates were concentrated at 60 °C and freeze-dried for 24 h. The main chemical ingredients of LPTE were characterized by UPLC-MS. As shown in Appendix A, the conditions used were the same as in our previous study [27]. The analysis was conducted using a Waters ACQUITY UPLC I-Class PLUS System (Waters, Milford, MA, USA) (SN: L22BSP766G), coupled with a SELECT SERIES Cyclic IMS (Waters, Milford, MA, USA) (SN: GBC086).

### 2.3. Chemometric Analysis of LPTE

In accordance with professional analytical standards, the major components of LPTE were quantified using chemometric analysis. Specifically, total polyphenols were determined by the Folin–Ciocalteu method (GB/T 8313-2018 [32]), total flavonoids by the sodium nitrite-aluminium nitrite method (SN/T 4592-2016 [33]), total free amino acids by the ninhydrin colorimetric method (GB/T 8314–2013 [34]), and total polysaccharides by the phenol-sulfuric acid method (NY/T 1676-2008 [35]). Additionally, total soluble sugars were quantified using a modified version of the anthrone-sulfuric acid method based on previous studies, ensuring consistency and accuracy in the analysis [36].

### 2.4. Network Pharmacology and Molecular Docking Analyses

This analytical approach was conducted in accordance with our previous research [27]. The molecular structures of quercetin, luteolin, ellagic acid, and kaempferol were retrieved from the PubChem database. The crystal structures of target proteins were obtained from the Protein Data Bank (PDB) as follows: PTGS2 (PDB ID: 5F19, 2.04 Å), CCNB1 (PDB ID: 6GU2, 2.00 Å), CCND1 (PDB ID: 6P8E, 2.30 Å), and EGFR (PDB ID: 8FV3, 2.10 Å). All protein structures underwent energy minimization using the Molecular Operating Environment (MOE) 2019 software prior to docking. Subsequently, molecular docking simulations were conducted using MOE software to evaluate the interaction strength between each compound and the corresponding protein binding sites.

### 2.5. Identification and Quantification of Potential Active Ingredients in LPTE

To identify the potential active compounds in LPTE, standard reference compounds were analyzed using the same UPLC-MS instrumentation (Waters, Milford, MA, USA) and analytical conditions as described in Section 2.2. Compounds were identified by comparing their retention times and characteristic fragment ions with those of the standards.

Quantitative analysis was performed using an Agilent 1260 HPLC System (Agilent, Santa Clara, CA, USA) equipped with a TC-C18 column (5 μm, 4.6 mm × 100 mm), coupled to a 6545 Q-TOF mass spectrometer. The mobile phase consisted of phase A (acetic acid/water, 1:1000, *v*/*v*) and phase B (acetonitrile). The flow rate was kept at 0.4 mL/min with a 20 μL injection volume at 30 °C. The elution gradient was 0–45 min, 20–70% mobile phase B; 45–50 min, 20% mobile phase B.

### 2.6. Animals and Management

All animal experiments were approved by the Animal Experimental Ethics Committee of Guangxi University (Approval No. GXU-2023-0225). Male DBA/1 mice (6 weeks old, 21 ± 2 g) were purchased from Hangzhou Ziyuan Laboratory Animal Technology Co., Ltd. (Hangzhou, China) (License No. 20230313Abbz0105000347). The animals were housed under controlled conditions with a temperature of 24 ± 2 °C, humidity at 55 ± 5%, and a 12 h light/dark cycle. All experimental procedures complied with the ethical standards for animal research and the ARRIVE guidelines.

After a one-week acclimation period, a total of 25 mice were randomly assigned to five groups (*n* = 5): normal group (healthy control), model group (CIA mouse model), PC group (positive control, MTX, 1 mg/kg/3 days), L group (low-dose LPTE, 200 mg/kg/day), and H group (high-dose LPTE, 600 mg/kg/day) [37,38]. MTX is a commonly used disease-modifying antirheumatic drug for the treatment of RA. CIA was induced in all groups except the normal group by subcutaneous injection of an emulsion of chicken type II collagen and complete Freund’s adjuvant at the base of the tail, followed by a booster injection with incomplete Freund’s adjuvant on day 21, according to the manufacturer’s protocol. RA articular damage scores and body weights were recorded every 3 days starting from day 21, with detailed scoring criteria provided in the Appendix A. The arthritis score for each mouse was calculated as the sum of scores from all four paws, with a maximum possible score of 16 [7]. From day 1 to day 72, the PC, L, and H groups received oral administration of MTX or LPTE at their respective doses, while the normal and model groups were given an equivalent volume of saline by gavage. On the final day, all animals were humanely euthanized, and samples of blood, paws, hind legs, and colon contents were collected. Their liver and spleen weights were also recorded for further analysis.

### 2.7. Micro-CT Imaging

Mouse hind paws were fixed in 4% paraformaldehyde for 48 h and scanned using high-resolution micro-CT (SkyScan, Kontich, Belgium). The scanning parameters were set as follows: voltage at 70 kV, electric current at 200 µA, exposure time at 460 ms, and scanning resolution at 10 μm. Bone mineral density (BMD, g cm^−3^), bone volume (BV, mm^3^), bone volume to tissue volume ratio (BV/TV, %), trabecular thickness (Tb.Th, mm), trabecular separation (Tb.Sp, mm), and trabecular number (Tb.N, mm^−1^) were analyzed using CT analysis software (CTAn 1.17, Kontich, Belgium) [7].

### 2.8. Histopathological Analysis

Paws were disarticulated from the skin following micro-CT imaging, decalcified in 5.5% disodium EDTA for 21 days, embedded in paraffin, sectioned into 5 μm slices, and stained with Safranin O/fast green as well as hematoxylin and eosin (H&E) [39]. Cartilage degradation and synovial inflammation in the knee and ankle joints were assessed by examining bone and synovial tissues under 200× magnification with an optical microscope (UOP, Chongqing, China).

### 2.9. Enzyme-Linked Immunosorbent Assay (ELISA)

IL-1β, TNF-α, IL-6, and IL-10 levels in serum were measured using ELISA kits (Jonln, Shanghai, China) according to the manufacturer’s instructions.

### 2.10. Serum Metabolomic Analysis

Serum samples were prepared using a method derived from our prior research, with tamoxifen and cholic acid-d4 as internal standards [27]. Briefly, serum samples were thawed at 4 °C, mixed with acetonitrile, and centrifuged. The supernatant was vacuum-dried, reconstituted with 75% acetonitrile containing 0.5 μM internal standards. The quality control sample comprised an equal volume of supernatant from all samples (6 μL). Serum metabolomic profiles were analyzed using a UPLC-MS system, with the LC and MS parameters utilized in the metabolomic investigation are detailed in Appendix A. The analysis was conducted using a Waters ACQUITY UPLC I-Class PLUS System (Waters, Milford, MA, USA) (SN: L22BSP766G) coupled to a SELECT SERIES Cyclic IMS (Milford, MA, USA) (SN: GBC086).

### 2.11. Determination of Alterations in Gene Expression Levels via qPCR

Total RNA was extracted from the paws, knee joint, and colon using TransZol Up reagent (TransGen Biotech, Shanghai, China), followed by cDNA synthesis with SweScript All-in-One RT SuperMix. Subsequently, PCR amplification was performed using MonAmp SYBR Green qPCR Mix [11]. *GADPH* was used as an internal reference, and the mRNA expression levels of *Alox5*, *Ptgs2*, *Cbr1*, *ZO-1*, *Occludin*, *GPR41*, and *GPR109a* genes were normalized to *GADPH* expression and analyzed using the 2^−ΔΔCT^ method. The gene primers were listed in Appendix A.

### 2.12. Gut Microbiota Analysis

Cecum content from each mouse was individually collected, labeled with a unique identifier, and then stored at −80 °C for subsequent analysis. The bacterial DNA extraction, PCR amplification, and sequencing were performed according to the standard protocol provided by Beijing Novogene Co., Ltd. (Beijing, China), as detailed in the referenced publication [27]. Detailed experimental methods and analyses were presented in the Appendix A.

### 2.13. Fecal SCFA Metabolomic Analysis

SCFAs from fecal samples were extracted, derivatized, and analyzed following an established method with minor modifications [40]. Procedures for SCFA preparation and GC/MS analysis parameters were outlined in the Appendix A. Quantification of SCFAs in the samples was achieved by referencing external calibration curves based on the peak areas in standard solutions.

### 2.14. Data Processing and Statistical Analysis

Metabolomic data were processed following our previous study [28]. The results were presented as mean ± standard deviation. Multivariate statistical analyses were performed using SIMCA-P 14.0 (Umetrics, Umea, Sweden). Statistical analysis of data was conducted with the Mann–Whitney *U* test and Student’s t-test. A statistical indicator of *p* < 0.05 was used to identify variables with significant differences between groups. Receiver operating characteristic (ROC) curve and pathway enrichment analyses were processed by the MetaboAnalyst 6.0 database. Spearman correlation analysis was performed to evaluate associations between variables. Additionally, the tracing analysis of differential metabolites was performed using MetOrigin (http://MetOrigin.met-bioinformatics.cn/).

## 3. Results and Discussion

### 3.1. Phytochemical Profile and Main Components Analyses of LPTE

The aqueous extracts of LPT were analyzed by LC-MS-based untargeted metabolomics, leading to the identification of 41 potential active components. Of these, 27 components were detected in positive ionization mode, and 14 components were identified in negative ionization mode (Appendix A, respectively). Notably, flavonoids were the predominant class of compounds identified. This comprehensive phytochemical profile provided a detailed characterization of LPTE and set the foundation for subsequent pharmacological studies to discover its active components.

Further chemometric analyses were performed to quantify the bioactive constituents of LPTE, focusing on compounds such as total flavonoids, total soluble sugars, polyphenols, polysaccharides, and free amino acids. Flavonoids inhibit inflammatory mediators and cytokines at the genetic and protein levels, modulate pro-inflammatory enzyme activity, and thereby exhibit anti-arthritic properties [41]. Polyphenols and polysaccharides have potential therapeutic effects in RA by modulating signaling pathways such as NF-κB and MAPKs [42,43]. Based on the current methods, we primarily measured the content of these compounds, thereby constructing the primary nutritional profile of LPTE. The results revealed that polyphenols and flavonoids predominated, with concentrations of 142.53 ± 0.98 mg/g and 93.05 ± 6.82 mg/g, respectively (Appendix A). Given their well-known anti-inflammatory and antioxidant properties, polyphenols and flavonoids are likely key contributors to the therapeutic potential of LPTE. To further explore this potential, a network pharmacology approach was employed to identify therapeutic targets and elucidate the mechanisms behind LPTE’s effects on RA.

### 3.2. Network Pharmacology Analysis

The screened compounds were imported into a pharmacological database, yielding 199 potential therapeutic targets. Concurrently, a total of 9989 RA-associated target genes were retrieved from the GEO dataset (GSE55235), and a Venn diagram revealed 139 overlapping targets (Figure 1A), which were considered potential therapeutic targets of LPTE in RA. Topological analysis of the protein–protein interaction network identified the top 10 key genes (Figure 1B), and KEGG pathway enrichment revealed significant associations with cancer, serotonergic synapse, and nitrogen metabolism pathways (Figure 1C). Notably, the arachidonic acid metabolism and FoxO signaling pathways, both strongly implicated in RA pathophysiology, were identified as the key pathways through which LPTE may exert its therapeutic effects. A network integrating LPTE core components, RA targets, and related pathways was constructed using Cytoscape software 3.9.1 (Figure 1D), showing the potential of LPTE to modulate these pathways, thus contributing to RA management. Furthermore, network pharmacology analysis identified four key bioactive compounds in LPTE (quercetin, luteolin, ellagic acid, and kaempferol) that potentially interact with core targets in the arachidonic acid metabolism and FoxO signaling pathways. Quercetin has improved RA symptoms significantly by reducing levels of pro-inflammatory cytokines such as TNF-α, IL-6, and IL-1β [44]. Ellagic acid could alleviate the severity of RA in CIA rats through downregulating MTA1/HDAC1 complex and promoting HDAC1 deacetylation-mediated Nur77 expression [45]. Luteolin exhibits anti-RA through inhibiting TLR4 signaling [46]. Kaempferol treats rheumatoid arthritis by modulating the NLRP3/CASP1/GSDMD axis and T cell activation [47]. These candidate compounds were subsequently verified using chemical standards based on the comparison of retention time, exact molecular weight, and MS/MS fragments (Appendix A).

### 3.3. Quantitative Analysis of Key Characteristic Compounds in LPTE

After identification using reference standards, we performed quantitative analysis of key characteristic compounds in LPTE, including caffeine, gallic acid, ellagic acid, quercetin, luteolin, and kaempferol, using the LC-MS technique. Caffeine emerged as the most abundant component (38.86 ± 2.18 mg/g), significantly surpassing levels of gallic acid (0.70 ± 0.13 mg/g), ellagic acid (0.64 ± 0.09 mg/g), quercetin (0.27 ± 0.05 mg/g), kaempferol (0.21 ± 0.02 mg/g), and luteolin (0.07 ± 0.02 mg/g) (Figure 2). Interestingly, while caffeine is recognized for its central nervous system activity, the fermentation process reduces its content in LPT by approximately 4.5% compared to non-fermented tea [48]. This compositional profile suggests LPTE may provide a favorable balance between bioactive benefits and tolerability.

### 3.4. Molecular Docking Analysis

To investigate the binding affinity of key compounds of LPTE to RA-related targets, molecular docking was performed. In the docking study, lower protein–ligand binding energy values indicated better binding, with negative binding free energy values suggesting a spontaneous binding process [49]. The docking results, visualized with MOE software (Appendix A) and supported by calculated binding energies (Table 1), demonstrated strong binding activity between the active compounds and RA target proteins. Among the target proteins, PTGS2 exhibited the highest binding affinity with quercetin, luteolin, ellagic acid, and kaempferol, with binding energy scores of −6.7242, −6.8201, −6.6828, and −6.6104, respectively. Various binding modes, including hydrogen bonds (H-bonds), H-π, and π-π interactions with key amino acid residues, were observed (Appendix A). These results suggest that the strong binding affinities of LPTE compounds with RA-related targets, especially PTGS2, may contribute to their therapeutic mechanisms. To evaluate whether these molecular interactions translate into in vivo efficacy, we proceeded to assess the therapeutic effects of LPTE in a RA animal model.

### 3.5. LPTE Attenuated the Severity of Arthritis and Bone Destruction in RA Mice

In this study, we examined the therapeutic effects of LPTE and explored its potential mechanisms using the CIA mouse model, which closely mimics the histopathological and immunological features of human RA [50]. As shown in Figure 3A, CIA resulted in significant paw swelling and redness, while management with LPTE and MTX alleviated these symptoms. Among LPTE-treated groups, the H group demonstrated more pronounced symptom relief compared to the L group, and the PC group (MTX) showed the strongest therapeutic effect. Arthritis severity was assessed using clinical arthritis scores recorded throughout the 72-day experimental period. As shown in Figure 3B, arthritis scores in the model group increased sharply after day 21, while significant inhibition of RA progression was observed in the high-dose LPTE and MTX groups after day 51 (*p* < 0.05). Following the second immunization, both the H and PC groups exhibited similar weight gain trends, which were significantly different from the model group (*p* < 0.05). No significant difference in weight gain was observed between the L group and the model group (Figure 3C).

Chronic inflammation and immune dysregulation are hallmarks of RA pathogenesis [51]. After euthanasia, liver and spleen weights of mice were measured to calculate the organ index (Figure 3D,E). The model group showed significantly increased liver and spleen weights, indicating the heightened immune activity (*p* < 0.05). However, management with high-dose LPTE reduced the liver index and significantly lowered the spleen index, suggesting that LPTE inhibited RA progression without inducing notable adverse effects (*p* < 0.05). During RA progression, pro-inflammatory cytokines such as IL-1β, TNF-α, and IL-6 drive the production of matrix metalloproteinases and promote immune cell infiltration into joint tissues, thereby exacerbating inflammation and accelerating joint destruction [52]. Conversely, IL-10, a key anti-inflammatory cytokine, is typically reduced in RA, contributing to immune imbalance and influencing the onset and progression of RA. Quantitative ELISA disclosed significantly elevated levels of pro-inflammatory cytokines (IL-1β, TNF-α, and IL-6) and significantly reduced levels of the anti-inflammatory cytokine IL-10 in the model group compared to the normal group (*p* < 0.05). Management with high-dose LPTE and MTX led to a significant decrease in pro-inflammatory cytokines (IL-1β, TNF-α, and IL-6) and a significant increase in anti-inflammatory cytokine IL-10 (*p* < 0.05) (Figure 3F–I).

Micro-CT analysis showed severe bone structure destruction in the paw joints of the model group, which was significantly alleviated by high-dose LPTE and MTX management (Figure 3J). Quantitative analysis of bone parameters revealed that high-dose LPTE and MTX management significantly improved BMD, BV, Tb.Th, Tb.N, BV/TV, and BV, while significantly reducing Tb.Sp compared to the model group (*p* < 0.05) (Figure 3K–P). Histological analysis of joint tissues revealed synovial hyperplasia, articular erosion, and inflammatory cell infiltration in the model group. These pathological changes were substantially alleviated in LPTE- and MTX-managed groups (Figure 3Q,S). Safranin O staining further confirmed the cartilage preservation in LPTE-managed mice, whereas the model group exhibited severe cartilage degradation (Figure 3R,T).

Collectively, these findings indicated that LPTE effectively prevented joint destruction, cartilage degradation, and inflammation in RA mice, with particularly strong therapeutic effects observed in the high-dose LPTE group. Given its promising in vivo efficacy, we further investigated LPTE-induced metabolic and microbial alterations to gain insights into its broader systemic effects.

### 3.6. Serum Metabolomic Analysis of LPTE in RA Management

Orthogonal partial least squares discriminant analysis (OPLS-DA) revealed clear serum metabolite profile distinctions among the groups, with strong separation between the normal and model groups (Figure 4A,B). The L group’s metabolic profile closely resembled that of the model group, while the H group’s profile was more similar to the normal group, indicating dose-dependent metabolic improvements following LPTE intervention. The OPLS-DA model was further used to identify differential metabolites between the model group and either the LPTE-managed or normal groups (Figure 4C–E). According to the criteria of VIP >1, *p* < 0.05, and FC ≥ 1.2 or FC ≤ 0.8, 117 differential metabolites were identified between the normal and model groups, and 44 differential metabolites were identified between the model and H groups, respectively (Appendix A). Shared differential metabolites between the normal vs. model groups and H vs. model groups were analyzed further. Notably, the metabolic profile of the H group closely aligned with that of the normal group (Appendix A). Key metabolites of arachidonic acid, including 8-HETE, 12(R)-HETrE, and 8-isoprostaglandin F2α, were identified as major contributors to RA pathogenesis and LPTE’s therapeutic effects, with significant upregulation of these metabolites in the model group, returning to levels similar to the normal group following LPTE intervention. 8-HETE has been shown to be significantly elevated in both the knee joints and plasma of RA, promoting leukocyte recruitment to the site of inflammation and their attachment to inflamed and damaged tissues, contributing to joint damage [53,54]. Inhibiting the production of 8-HETE can improve metabolic dysregulation and help restore the pathological processes of RA [54]. Likewise, 12(R)-HETrE exhibits potent pro-inflammatory and angiogenic activities, exacerbating vascular inflammation in RA progression [55]. Additionally, inflammatory mediators produced by chondrocytes in RA stimulate the release of 8-isoprostaglandin F2α, which exacerbates cartilage degradation and bone damage, while relevant receptor antagonism significantly lowers its levels in chondrocytes, thereby reducing joint inflammation [56]. Receiver operating characteristic curve (ROC) analysis identified five key differential metabolites, including 1-stearoylglycerophosphoinositol, 8-HETE, tetrahydrocortisone, LPE (20:5), and retinyl beta-glucuronide, showing excellent diagnostic accuracy (AUC = 1), marking them as potential biomarkers for both RA diagnosis and LPTE efficacy (Appendix A).

Pie chart analyses revealed that fatty acids, glycerophospholipids, and prenol lipids serve as the primary differential metabolites distinguishing both the normal vs. model groups and the model vs. H groups. Notably, fatty acids accounted for the highest proportion of differential metabolites in both comparisons (Figure 5A,B). Pathway enrichment analysis was performed using MetaboAnalyst. Differential metabolites between the normal and model groups were significantly enriched in linoleic acid, arachidonic acid, and steroid hormone biosynthesis metabolism (*p* < 0.05). Notably, arachidonic acid metabolism was also markedly enriched in the model vs. H groups comparison (*p* < 0.05) (Figure 5C,D), underscoring its critical role in the therapeutic effects of LPTE. Arachidonic acid, a key mediator of inflammation, facilitates the production of inflammatory cytokines and acts as a physiological trigger for apoptosis. Its pro-inflammatory and oxidative stress-related properties contribute to necrotic and pathological conditions associated with RA [57]. To further explore the involvement of arachidonic acid metabolism, the expression of key genes (*Alox5*, *Ptgs2*, and *Cbr1*) was analyzed in paw tissues. In the model group, the expressions of *Alox5*, *Ptgs2*, and *Cbr1* were significantly upregulated compared to the normal group (*p* < 0.05). The expression patterns of *Alox5* and *Ptgs2* were consistent with human synovial tissue data from the GSE77298 dataset. In particular, *Ptgs2*, which encodes COX-2, is a pivotal enzyme in pro-inflammatory prostaglandin synthesis and is typically overexpressed in RA synovial fibroblasts, underscoring its significance as a therapeutic target [58,6]. LPTE management significantly downregulated the overexpression of these genes, further supporting its role in modulating arachidonic acid metabolism (*p* < 0.05) (Figure 5E–G). Combined with findings from network pharmacology analysis, these results strongly suggest that LPTE alleviated RA by modulating arachidonic acid metabolism.

To further explore the metabolic origin of LPTE’s effects, metabolite traceability and metabolic function analyses were conducted using the MetOrigin platform. In the comparison between model and H groups, 12 bacteria–host cometabolites and 1 host-specific metabolite were identified as the key contributors to LPTE-mediated efficacy (Figure 5I,K). Metabolite pathway enrichment analysis (MPEA) revealed obvious regulation of one host-specific and six cometabolism pathways, with arachidonic acid and glycerophospholipid metabolism displaying the most prominent alterations (Figure 5H,J). In the comparison between normal and model groups, 26 bacteria–host cometabolites, 3 bacterial metabolites, and 4 host-specific metabolites were identified, respectively (Appendix A). MPEA analysis further indicated that sphingolipid, linoleic acid, arachidonic acid, and glycerophospholipid metabolism were key pathways significantly influenced by gut microbiota during RA progression (Appendix A). These findings underscore the critical role of gut microbes in shaping host metabolism, particularly in the context of RA pathogenesis and management. To further define the contribution of specific microbial species in mediating LPTE’s effects, subsequent analysis focused on changes in microbial community composition in response to LPTE intervention.

### 3.7. LPTE Reshaped the Gut Microbiota Profiles in RA Mice

Gut microbiota dysbiosis plays a pivotal role in the progression of RA, often exacerbating systemic inflammation [59]. In this study, we assessed the impact of LPTE management on the gut microbiota of RA mice by analyzing the cecum contents. Alpha diversity indices, including Observed_features and Chao1 (Figure 6A,B), revealed a reduction in amplicon sequence variants (ASVs) and microbial richness in the model group compared to the normal group. However, LPTE management, particularly at high doses (H group), demonstrated a restoring effect on microbial richness. For beta diversity analysis, a PLS-DA model evaluated the microbial community composition. The results showed distinct separation between the normal and model groups, underscoring gut microbiota dysbiosis in RA (Figure 6C). Following LPTE management, the microbial community structure shifted obviously, with the microbiota profile of the H group closely resembling that of the normal group, suggesting that LPTE modulates gut microbial composition in RA mice.

To further explore these changes, the overall microbial composition was analyzed at the class and family levels. At the class level, *Clostridia*, *Bacteroidia*, and *Bacilli* were the predominant in all groups. The RA model group exhibited dysbiosis, characterized by reduced abundances of *Clostridia* and *Bacteroidia* and an increased abundance of *Bacilli*. LPTE management reversed these changes, restoring the levels of *Clostridia* and *Bacteroidia* while reducing *Bacilli* abundance (Figure 6D). At the family level, *Lachnospiraceae*, *Lactobacillaceae*, and *Muribaculaceae* dominated all groups (Figure 6E). Specifically, the model group also exhibited dysbiosis, with an increased abundance of *Lactobacillaceae*, *Oscillospiraceae*, and *Rikenellaceae*, along with decreased abundance of *Lachnospiraceae*, *Muribaculaceae*, *Desulfovibrionaceae*, *Prevotellaceae*, *Ruminococcaceae*, *Saccharimonadaceae*, and *Eggerthellaceae*. High-dose LPTE management effectively reversed these trends by restoring bacterial abundances and composition closer to normal levels, notably restoring the abundance of beneficial families. Moreover, a clustering heatmap based on the top 35 ranked families provided additional support for the LPTE’s improvement in gut microbiota (Figure 6F). These findings collectively suggested that LPTE effectively restored gut microbiota composition and balance in RA mice. To further elucidate the role of these microbial alterations, we next focused on the SCFA and gut barrier integrity as key factors contributing to LPTE’s therapeutic effects.

### 3.8. LPTE Enhanced Intestinal Barrier Function and Increased SCFAs Production

SCFAs, key metabolites produced by gut microbiota fermentation of dietary fiber in the gastrointestinal tract, act as a critical link between gut microbiota and immune system regulation. SCFAs possess potent immunomodulatory properties, modulating inflammatory processes and acting as essential mediators of gut microbiota’s systemic effects [60]. To evaluate LPTE’s effects on SCFA production, we measured SCFA levels in fecal samples from mice. Compared to the normal group, the SCFA levels in the model group were significantly reduced, with propionic acid, butyric acid, and isovaleric acid showing the most pronounced decreases (*p* < 0.05). After LPTE management, SCFA levels were significantly restored in both the H and L groups, especially for acetic acid, propionic acid, and butyric acid, which were elevated to levels comparable to those in the normal group (*p* < 0.05) (Figure 7A–H). These results indicated that LPTE intervention effectively restored SCFA levels in RA mice. SCFAs are known to be natural ligands for G protein-coupled receptors (GPCRs) and activate GPR41 and GPR109a to mediate their effects, with GPR41 activated by acetic, propionic, and butyric acids, and GPR109a activated by butyric and nicotinic acids [61,62]. Therefore, we measured mRNA expressions of SCFA receptors, including *GPR109a* and *GPR41,* in the colon and joint. Gene expression levels of *GPR109a* and *GPR41* were significantly upregulated in RA mice managed with high-dose LPTE (*p* < 0.05) (Figure 7I–L), suggesting the involvement of SCFAs in LPTE-mediated RA mitigation.

Gut microbiota and its derived SCFAs play crucial roles in maintaining intestinal barrier integrity [17]. The intestinal barrier is primarily regulated by tight junction proteins, including Occludin and zonula occludens-1 (ZO-1) [63]. The model group exhibited significantly lower mRNA expression of these proteins compared to the normal group (*p* < 0.05). Whereas LPTE management, particularly at the high dose, markedly increased the expression of these tight junction proteins (Figure 7M,N), suggesting a restoration of intestinal barrier integrity (*p* < 0.05). These findings demonstrated that LPTE enhanced intestinal health by promoting SCFA production and increasing the expression of tight junction proteins, ultimately counteracting gut dysbiosis and reinforcing the intestinal barrier in RA mice. Given the interplay between gut microbiota, SCFAs, and host immunity, we next investigated their potential relationships with RA-related clinical parameters using integrated correlation analysis.

### 3.9. Integrated Correlation Analysis of Gut Microbiota, Differential Metabolites, SCFAs, and RA-Related Factors

Spearman correlation analysis showed that several differential metabolites, such as 8-HETE, 12(R)-HETrE, and 8-Isoprostaglandin F2a, which are metabolites of arachidonic acid, were significantly positively correlated with *Monoglobaceae* (*p* < 0.05) (Appendix A). These metabolites were positively associated with Tb.Sp, while demonstrating negative correlations with beneficial clinical factors such as BMD, Tb.Th, and BV/TV (Appendix A). The composition of the gut microbiota was also closely correlated with RA clinical outcomes. For instance, *Desulfovibrionaceae* and *Eggerthellaceae* were positively correlated with beneficial bone factors, whereas *Monoglobaceae* was negatively correlated with BMD, potentially influencing RA progression (Appendix A). *Desulfovibrionaceae* contributes to acetate production, supporting SCFA-mediated anti-inflammatory effects [64]. *Eggerthellaceae* metabolizes dietary polyphenols into bioactive phenolic compounds, which enhance lipid metabolism and promote hepatic detoxification, thereby contributing to systemic immune modulation in mice [65]. *Monoglobaceae* displayed a positive correlation with the expression of key immune-related genes such as *Tlr7*, *Tlr12*, and *LBP*, which are involved in immune response regulation in the mouse colon [66]. TLRs mediate pathogen recognition and trigger inflammatory signaling cascades, which are critical for innate immune activation [67]. Consequently, *Desulfovibrionaceae* and *Eggerthellaceae* may contribute to the therapeutic effects of LPTE, while certain metabolites such as 8-HETE, 12(R)-HETrE, and 8-Isoprostaglandin F2a, and gut microbiota (including *Monoglobaceae*), may promote RA progression.

Further correlation analysis revealed that SCFAs were positively correlated with beneficial clinical factors, emphasizing their potential role in RA management (Appendix A). Notably, isovaleric acid displayed a strong positive association with Tb.Th and BV/TV, while showing a significant negative correlation with Tb.Sp (*p* < 0.05). Isovaleric acid protects bone integrity by inhibiting the differentiation of bone marrow macrophages into osteoclasts [68]. These findings highlighted that isovaleric acid plays a crucial role in maintaining subchondral bone integrity (Appendix A). Analysis of SCFA and gut microbiota interactions revealed that 20 bacterial families correlated with SCFA levels. The abundance of *Eggerthellaceae* was positively correlated with the production of SCFAs, especially acetic acid, isobutyric acid, and isovaleric acid (*p* < 0.05). In contrast, *Monoglobaceae* exhibited a negative correlation with SCFA production, particularly propionic acid (*p* < 0.05) (Appendix A). In summary, modulation of specific bacterial families, such as *Eggerthellaceae* and *Monoglobaceae*, may influence SCFA biosynthesis, thereby improving bone health and enhancing the therapeutic effects of LPTE against RA.

The integrated correlation analysis revealed complex interactions between serum metabolites, gut microbiota profiles, SCFA levels, and RA-related bone parameters. These findings highlight the microbiota-mediated effects of LPTE and suggest its therapeutic benefits may arise through a multi-target mechanism involving metabolic regulation, gut microbiota modulation, and immune function adjustment.

## 4. Conclusions

To our knowledge, this is the first study to systematically evaluate the anti-RA potential of LPTE. LPTE effectively alleviated RA by regulating the gut–joint axis mediated by fatty acid metabolism. It regulates arachidonic acid metabolism by modulating key genes, including *Alox5*, *Ptgs2*, and *Cbr1*, thereby reducing joint inflammation of RA. Concurrently, LPTE reshaped the gut microbiota profiles, particularly enriching *Monoglobaceae*, *Eggerthellaceae*, and *Desulfovibrionaceae*, which promoted the production of SCFAs. Elevated SCFA levels enhanced intestinal barrier integrity and suppressed systemic inflammation by upregulating tight junction proteins (ZO-1 and Occludin) and activating SCFA receptor (GPR41 and GPR109a) signaling pathway. Through this dual regulation of fatty acid metabolism and gut microbiota, LPTE restored immune and microbial homeostasis, ultimately mitigating RA pathology. Preliminary evidence suggests that the observed anti-RA properties of LPTE may originate from its constituents, including quercetin, ellagic acid, luteolin, and kaempferol. These findings highlight the potential of LPT as a natural product and functional food for mitigating RA, suggesting its application as a health-promoting beverage for the prevention and management of RA. Future work will aim to clarify the individual contributions of the main bioactive compounds to the therapeutic effects of LPTE against RA (Figure 8).

## Figures and Tables

**Figure 1 foods-14-02854-f001:**
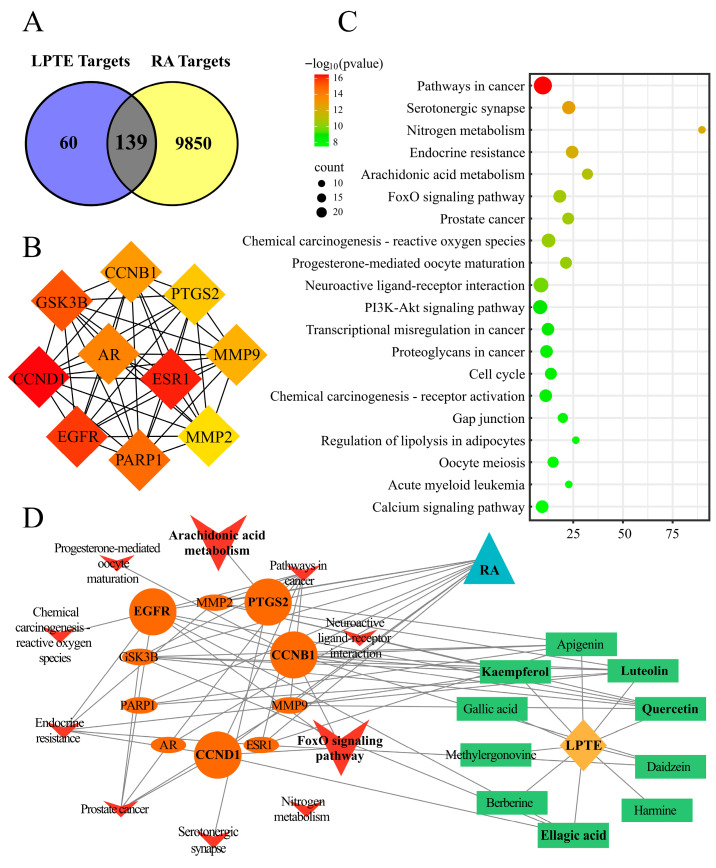
Network pharmacology analysis. (**A**) Venn diagram of LPTE targets and RA targets. (**B**) Top 10 key targets. (**C**) KEGG pathway enrichment analysis of 139 protein targets of LPTE in RA (top 20 were listed). (**D**) Pathway network of the core components of LPTE and the core targets of RA. LPTE, Liupao tea extract. RA, rheumatoid arthritis.

**Figure 2 foods-14-02854-f002:**
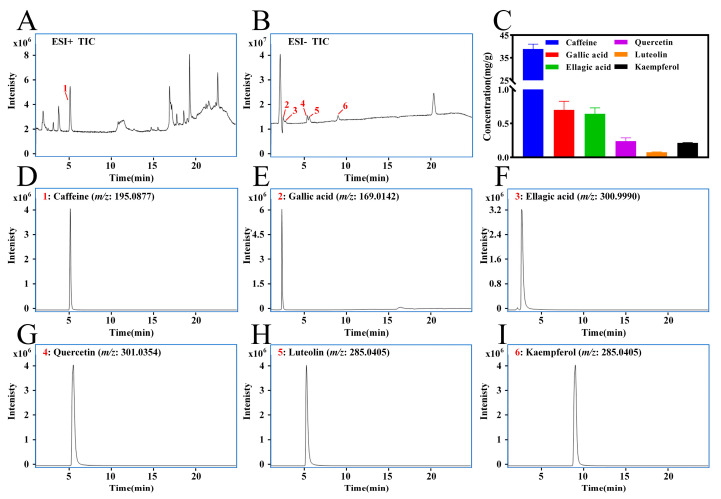
Quantitative analysis of key characteristic compounds. (**A**,**B**) Total ion chromatograms of LPTE in positive and negative modes. (**C**) Contents of caffeine, gallic acid, ellagic acid, quercetin, luteolin, and kaempferol in LPTE. (**D**–**I**) Mass spectra of caffeine, gallic acid, ellagic acid, quercetin, luteolin, and kaempferol, respectively.

**Figure 3 foods-14-02854-f003:**
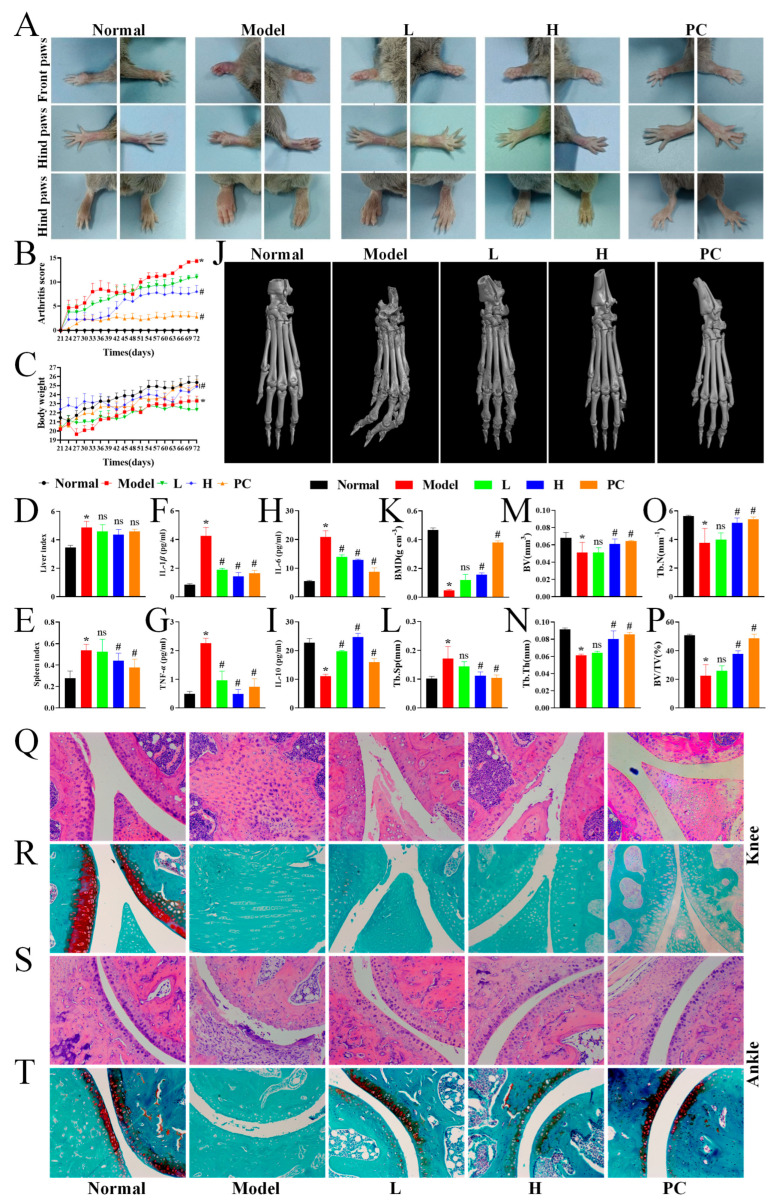
LPTE management attenuated arthritis symptoms. (**A**) Representative paw images of mice in different groups. (**B**) Arthritis score of four paws per mouse. (**C**) Body weight. (**D**,**E**) Index of liver and spleen. (**F**–**I**) Levels of IL-1β, TNF-α, IL-6, and IL-10. (**J**) Micro-CT 3D images of the hind paws of RA mice. (**K**) Bone mineral density (BMD, g cm^−3^). (**L**) Trabecular separation (Tb.Sp, mm). (**M**) Bone volume (BV, mm^3^). (**N**) Trabecular thickness (Tb.Th, mm). (**O**) Trabecular number (Tb.N, mm^−1^). (**P**) Bone volume to tissue volume ratio (BV/TV, %). (**Q**) H&E staining and (**R**) Safranin O/fast green staining of the knee. (**S**) H&E staining and (**T**) Safranin O/fast green staining of the ankle. Representative (**Q**–**T**) staining sections of the knee and ankle joint (200× magnification). Compared with the normal group, * *p* < 0.05; compared with the model group, # *p* < 0.05, ^ns^
*p* > 0.05.

**Figure 4 foods-14-02854-f004:**
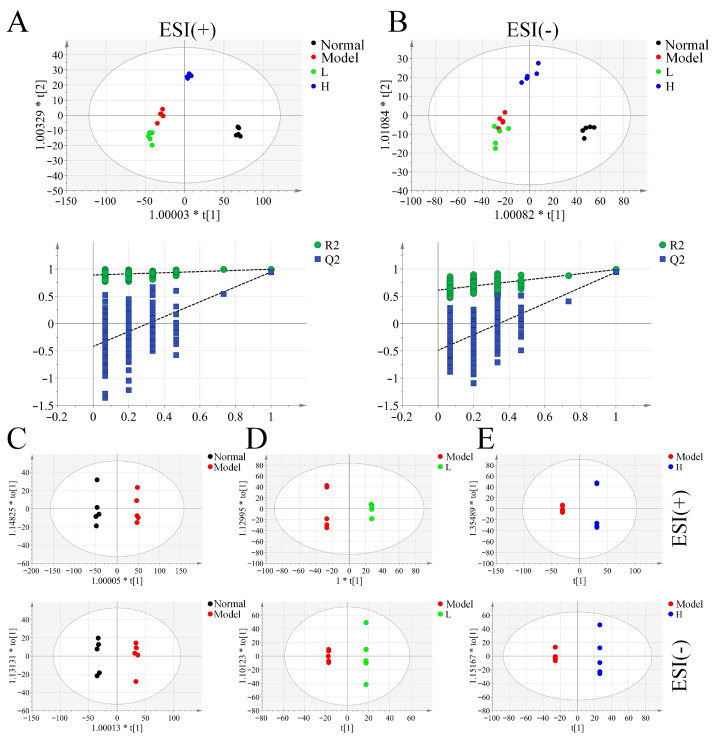
Multivariate statistical analysis of serum metabolomics (*n* = 5). (**A**,**B**) OPLS-DA score plots in ESI (+) and ESI (−) modes among four groups. (**C**–**E**) OPLS-DA score plots of different groups in both ESI modes. ESI, electrospray ionization. * denotes the scaling factor for variables.

**Figure 5 foods-14-02854-f005:**
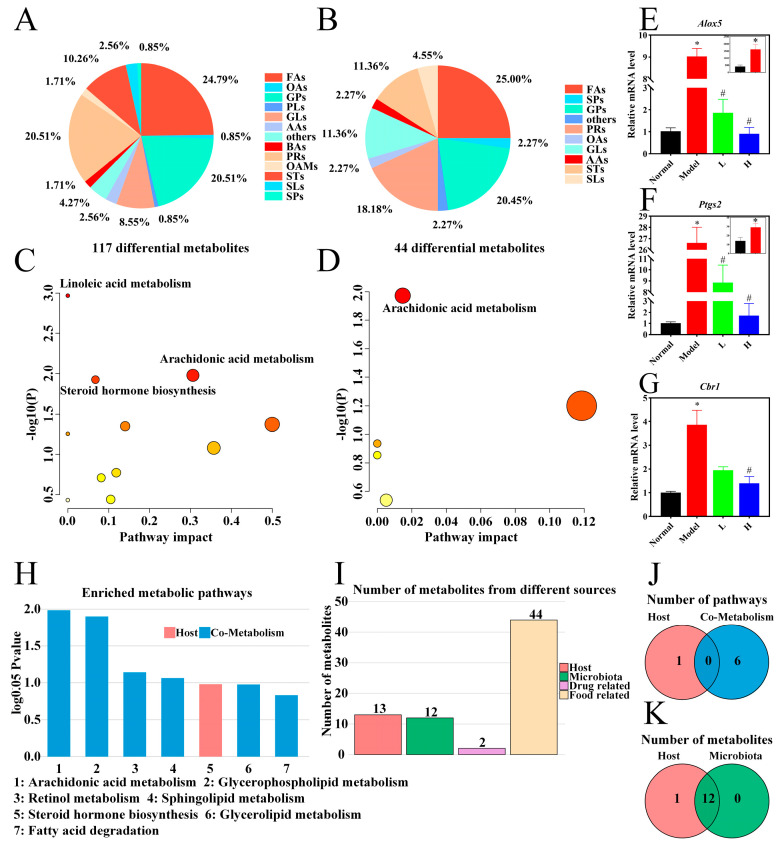
Influence of LPTE on the serum profiles in RA mice. (**A**–**D**) Composition and pathway analysis of differential metabolites in serum metabolomics (A and C: normal vs. model; B and D: model vs. H). (**E**–**G**) mRNA expression of *Alox5*, *Ptgs2*, and *Cbr1* in paw. (**H**,**J**) Histogram and Venn diagram of enrichment analysis of differential metabolic pathways (model vs. H). (**I**,**K**) Tracing the histogram and Venn diagram of differential metabolites between the model vs. H groups based on the MetOrigin platform. Compared with the normal group, * *p* < 0.05; compared with the model group, # *p* < 0.05.

**Figure 6 foods-14-02854-f006:**
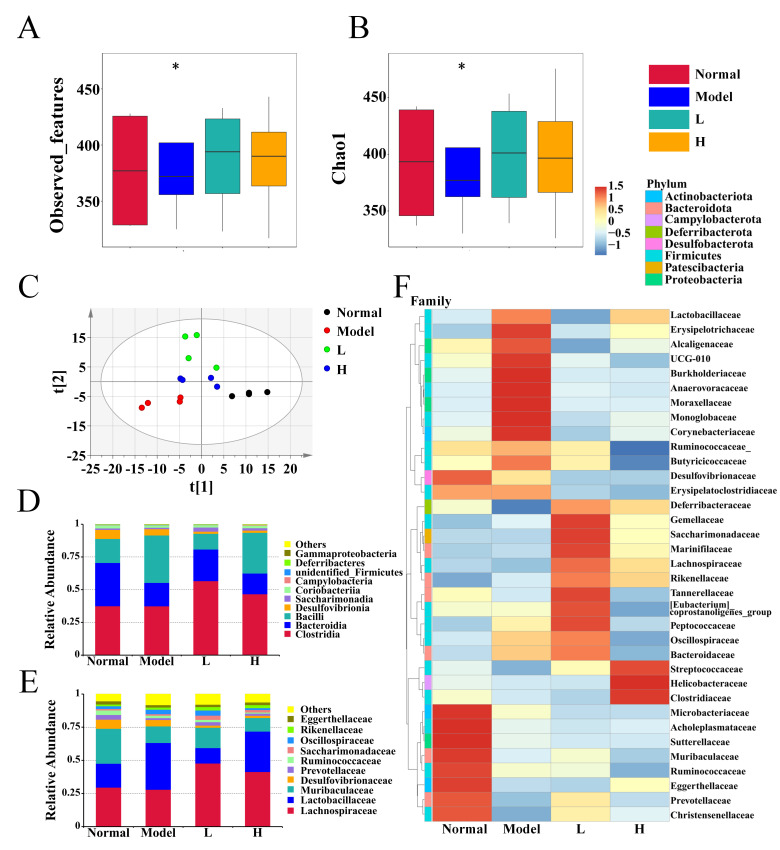
LPTE reshaped the microbial composition and alleviated its dysregulation in RA mice (*n* = 4). (**A**,**B**) α-diversity of gut microbiota by Observed_features and the Chao 1 index. (**C**) β-diversity of gut microbiota by partial least squares discriminant analysis (PLS-DA). (**D**,**E**) Relative abundance of gut microbiota at the class and family levels, respectively. (**F**) Heatmap of species abundance clustering of gut microbiota at the family level. Compared with the normal group, * *p* < 0.05.

**Figure 7 foods-14-02854-f007:**
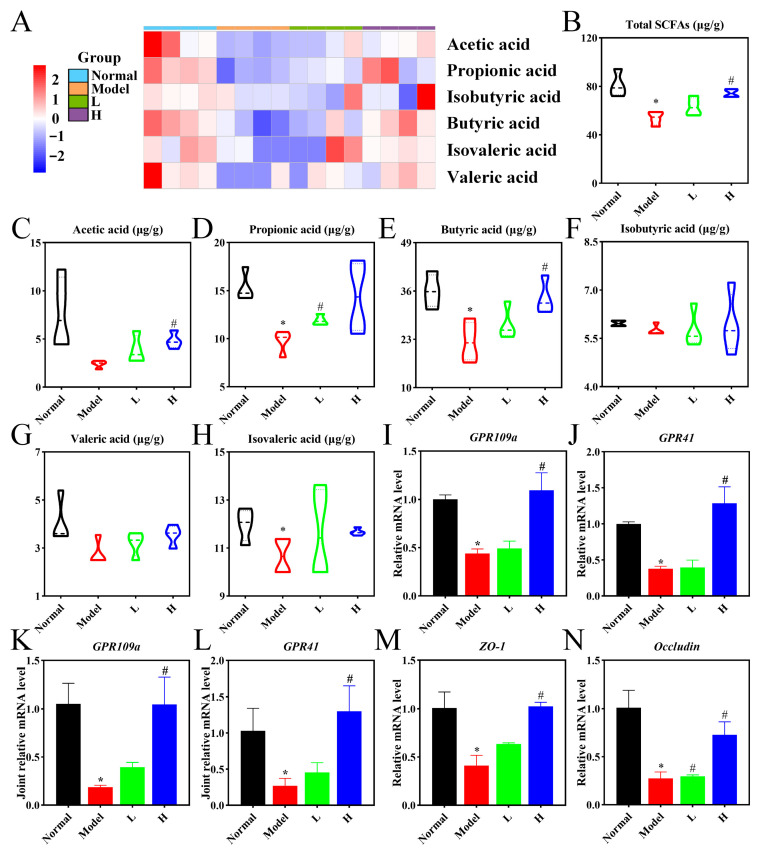
LPTE management restored the intestinal barrier, improved SCFA profiles, and activated SCFA receptors in RA mice. (**A**) Heatmap of SCFAs. The blue indicates lower abundance, and red indicates higher abundance. (**B**–**H**) Concentration of SCFAs, including total SCFAs, acetic acid, propionic acid, butyric acid, isobutyric acid, valeric acid, and isovaleric acid, from feces. (**I**,**J**) mRNA expression of *GPR109a* and *GPR41* in colon. (**K**,**L**) mRNA expression of *GPR109a* and *GPR41* in the knee joint. (**M**,**N**) mRNA expression of *ZO-1* and *Occludin* in the colon. Compared with the normal group, * *p* < 0.05; compared with the model group, # *p* < 0.05.

**Figure 8 foods-14-02854-f008:**
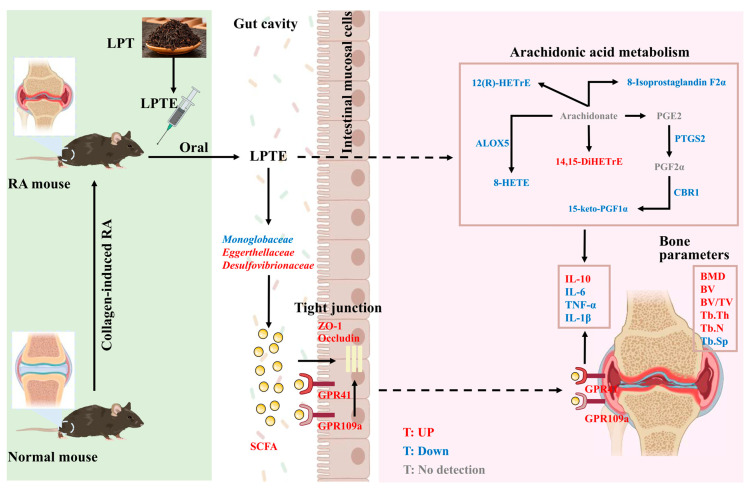
Schematic diagram of the mechanisms by which LPTE alleviates RA.

**Table 1 foods-14-02854-t001:** Interaction between key target genes of RA and core active ingredients of LPTE.

Gene Name (PDB ID)		Compound (Pubchem CID)			
		Quercetin (5280343)	Luteolin (5280445)	Ellagic Acid (5281855)	Kaempferol (5280863)
PTGS2 (5F19)	S score	−6.7242	−6.8201	−6.6828	−6.6104
	H-Bonds Type Amino acid	H-donor CYS 41 H-donor GLU 465 H-acceptor HIS 39	H-donor HIS 39 H-acceptor TYR 130	H-donor OAS 530	H-donor HIS 39 H-acceptor TYR 130
	π-Interactions Type Amino acid	pi-H CYS 47		pi-H VAL 523	
				pi-H ALA 527	
CCNB1 (6GU2)	S score	−6.0425	−6.0085	−5.8914	−5.8441
	H-Bonds Type Amino acid	H-donor ASP 146 H-donor ASP 128	H-donor GLU 51	H-acceptor LEU 83	H-donor ASP 146 H-donor ASP 128
	π-Interactions Type Amino acid			pi-H VAL 18	
CCND1 (6P8E)	S score	−5.4586	−5.5046	−5.0949	−5.4823
	H-Bonds Type Amino acid			H-donor GLY 228 H-acceptor ASN 276	
	π-Interactions Type Amino acid		pi-H LYS 152 pi-pi HIS 30		pi-H LYS 152 pi-pi HIS 30
EGFR (8FV3)	S score	−6.1760	−5.6858	−5.6894	−5.7273
	H-Bonds Type Amino acid	H-donor ASN 842 H-donor SER 720	H-donor ASN 842 H-donor ASP 855	H-acceptor MET 793	H-donor GLU 758 H-donor GLU 749
	π-Interactions Type Amino acid	pi-H VAL 726	pi-H VAL 726	pi-H LEU 718 pi-H VAL 726	pi-H LEU 861

## Data Availability

The original contributions presented in this study are included in the article/Appendix A. Further inquiries can be directed to the corresponding authors.

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
