# Peer review of "Liupao Tea Extract Alleviates Rheumatoid Arthritis in Mice by Regulating the Gut–Joint Axis Mediated via Fatty Acid Metabolism"

_foods, 2025, doi:10.3390/foods14162854_

Round 1
Reviewer 1 Report
Comments and Suggestions for Authors
The study looks at the effect of consuming Liupao tea in a mouse model of rheumatoid arthritis. The study addresses inflammatory and histological parameters and those related to the intestinal microbiota. The work is well structured and written, but some points, especially in the methodology, need to be improved.
- In the title of the paper and at various points in the text, the authors use the term “gut-fatty acid-joint axis”. However, I believe that the more appropriate term is gut-joint axis since the term refers to the interconnected relationship between the gut microbiome, its metabolites (particularly short-chain fatty acids or SCFAs), and the health of joints, including conditions like osteoarthritis and rheumatoid arthritis.
- In the introduction (lines 105-119) the authors give a summary of the main parameters that will be evaluated in the work. I suggest removing this part and adding a justification for the work.
- In terms of methodology, I suggest describing the methods used better. The authors do indeed have works with methodologies similar to those used in this work, but the methodology should be described minimally.
- Lines 145-151. "Chemometric analysis of LPTE" These analyses are of total phenolic compounds using the Folin? Describe all the methodologies or at least mention which methods were used.
- In item 2.4. describe at least which programs were used to do the docking.
- Sections 2.2, 2.5 and 2.6 describe the methods for characterizing and quantifying certain compounds in LPTE. I suggest putting all these items together or at least keeping them close. It is confusing for the reader the way it is presented. In addition, there is contradictory information about the equipment used, such as the type of HPLC and spectroscopy.
- In the methodology that describes the rheumatoid arthritis model, the model must be described. It is only cited as a CIA.
- In item 2.7. describe what “MTX” is
- The legend for Figure 2 must contain all the information in the figure. You can't understand the figure with the caption it has.
Reviewer 2 Report
Comments and Suggestions for Authors
Liupao Tea Extract Alleviates Rheumatoid Arthritis in Mice by Modulating the Gut-Fatty Acid-Joint Axis
- The author is suggested to improve the quality of abstract
- The author is suggested to give clarity on the reason for the people consuming this Liupao tea in china.
- If they are traditionally taking this Liupao tea, what is the purpose of this research – suggested for the clarification
- The author is suggested for clarification that, what is meant by collagen induced arthritis.
- The author is suggested to give 2-3 lines on the negative side of the Liupao tea consumption.
- The author is suggested to cross check the keywords
Introduction
- The author is suggested to include the reason for the condition of Rheumatoid Arthritis
- The author is suggested to give clarity, what exactly it means the pathological bone loss in RA
- The author is suggested to give the supporting literature references from 105-122.
Materials and Methods
- The author is suggested to reduce the 2.1. Materials and reagents
- The author is suggested to give the specification of UPLC-MS
- The author is suggested to give reference for the section, 2.11. Serum metabolomic analysis.
- The author is suggested to give reference for 2.12. Determination of alterations in gene expression levels via qPCR.
- The author is suggested to cross check the complete materials and methods section.
Results and Discussion:
- The authors were suggested to include the references in 3.4. Molecular docking analysis
- The author is suggested to include the supporting literature in the section 3.5
- The author is suggested to include the supporting literature in the session LPTE reshaped the gut microbiota profiles in RA mice.
- The author is suggested to cross verify the complete results and discussion part
- The author is suggested to include the supporting literature
References:
- The author is suggested to replace the old references with the newer ones
Reviewer 3 Report
Comments and Suggestions for Authors
The article investigates the effects of Liupao tea extract (LPTE) on rheumatoid arthritis in a mouse model. Using multi-omics approaches (metabolomics, microbiota analysis, qPCR), the study shows that LPTE alleviates joint inflammation and improves bone structure. These effects are linked to modulation of gut microbiota, intestinal barrier integrity, and fatty acid metabolism. Key active compounds identified include quercetin, luteolin, ellagic acid, and kaempferol.
While the study provides valuable insights at both the molecular and phenotypic levels, the causal relationship between these two dimensions remains somewhat speculative. Clearer evidence linking the molecular changes to the observed therapeutic effects would significantly strengthen the conclusions.
For instance, although the study reports changes in arachidonic acid metabolites, it does not directly demonstrate that these molecular alterations are responsible for the observed improvements in joint histology or bone microarchitecture. At a minimum, the authors should better support these findings with relevant references from the literature.
In addition, the manuscript would benefit from careful revision of the English language. Several sections suffer from grammatical inaccuracies, awkward phrasing, and inconsistent terminology, which sometimes hinder comprehension. A thorough proofreading is strongly recommended to improve clarity and ensure the scientific content is effectively communicated.
Below are my detailed comments:
-
Line 34: Instead of “joint-protective anti-inflammatory effects”, I suggest “joint-protective and anti-inflammatory effects”.
-
Line 36: Change “reduced pro-inflammatory cytokines levels” to “reduced the levels of pro-inflammatory cytokines”.
-
Line 49: The phrase “RA is significantly increased risk” should be revised to “RA is associated with a significantly increased risk of…”.
-
Lines 57-59: This sentence needs to be reformulated for clarity.
-
Lines 100-102: Please revise this sentence; it is syntactically incorrect and hard to follow.
-
Line 137: The authors mention using "reported methods with some modifications [28]", but it is unclear what modifications were made compared to the original method. Please specify the changes.
-
Figures 2 and 3: The resolution is low. Please provide higher-resolution images.
-
Line 585: What is meant by “reporter”? If referring to receptors like GPR41 and GPR109a, please use “receptor” instead.
-
Supplementary material: “Podofilox” should be corrected to “podophyllotoxin”.
-
Tables S6-S7: Were all the listed compounds tentatively identified, or were some confirmed using standard compounds? Please clarify.
-
Major active compounds: Why were the main compounds not tested individually? According to the quantitative analysis, the main flavonoids represent less than 1% of the extract. How can it be confirmed that they are responsible for the observed biological activity?
-
Line 259: Please revise “Flavonoids perfect the inhibition…” the phrasing is unclear and ungrammatical.
-
Line 253: Rephrase to “Flavonoids were the predominant class of compounds identified.”
-
Dosing rationale: How were the doses of 200 and 600 mg/kg/day selected? Please provide justification, ideally with references or preliminary data.
Round 2
Reviewer 1 Report
Comments and Suggestions for Authors
All suggested changes have been made.
Author Response
We greatly appreciate the reviewer’s positive feedback and support.
Reviewer 3 Report
Comments and Suggestions for Authors
The authors have addressed most of the issues. However, I suggest revising the manuscript before publication, especially to ensure the correct terminology is used. For example, "Inhibiting 8-HETE" is not appropriate, as HETE is not an enzyme; it would be better to say "inhibiting or reducing its production" instead.
Author Response
We sincerely appreciate the reviewer’s insightful suggestion regarding the proper use of terminology. In accordance with the comment, we have revised the phrase “inhibiting 8-HETE” to “inhibiting the production of 8-HETE” in the revised manuscript (line 411). This correction ensures terminological accuracy, as 8-HETE is a metabolite rather than an enzyme. Additionally, we conducted a comprehensive review of the manuscript to identify and correct any other potential misuses of terminology, ensuring consistency throughout. Furthermore, we have made improvements to grammar and wording to enhance clarity and readability. All changes made during this revision are marked in yellow highlight in the revised version for clarity. Thank you again for your valuable feedback, which significantly enhanced the scientific rigor of our manuscript.